# Modeling of Compressive Strength for Unidirectional Fiber Reinforced Composites with Nanoparticle Modified Epoxy Matrix

**DOI:** 10.3390/ma12233897

**Published:** 2019-11-26

**Authors:** Wei Chen, Yiping Liu, Zhenyu Jiang, Liqun Tang, Zejia Liu, Licheng Zhou

**Affiliations:** School of Civil Engineering and Transportation, State Key Laboratory of Subtropical Building Science, South China University of Technology, Guangzhou 510640, China; chenwei65@midea.com (W.C.); lqtang@scut.edu.cn (L.T.); zjliu@scut.edu.cn (Z.L.); ctlczhou@scut.edu.cn (L.Z.)

**Keywords:** fiber reinforced polymers, nanoparticles, compressive strength, damage, modeling

## Abstract

Incorporation of nanoparticles into polymer matrix was found to considerably improve the compressive performance of unidirectional fiber reinforced composites. In our experimental study, an increase by 62.7% in the longitudinal compressive strength of unidirectional carbon fiber reinforced composites is attained by dispersing 8.7 vol.% SiO_2_ nanoparticles into epoxy matrix. A compressive strength model is established to quantitatively describe the reinforcing effects of nanoparticles, which combines a modified microbuckling model for unidirectional fiber reinforced composites and a constitutive model for nanocomposite matrices under compression. In the two models, the coupling of damage and plasticity is considered to contribute to the nonlinear response of nanocomposite matrix. The proposed strength model demonstrates excellent prediction capability in experimental verification. A small relative deviation below 8.2% is achieved between the predicted compressive strength of unidirectional fiber reinforced composites and the measured values, which is at the same level of random error in experiments.

## 1. Introduction

Unidirectional fiber reinforced composites are generally designed to carry the tensile load along the reinforcing fibers. They show substantially inferior mechanical performances under longitudinal compression, due to the shape instability of fibers with high aspect ratio. The compressive strength has been found about 30%–40% lower than the tensile strength [1]. Unfortunately, longitudinal compression is almost inevitable for composite structures in engineering applications, which may experience relatively complex deformation (e.g., conditions including flexural deformation [2]). Thus, accurate prediction of compressive strength for fiber reinforced composites is a critical issue for the safe and optimal design of composite parts. It is known that during the compression of unidirectional fiber reinforced composites, microbuckling of fibers occurs at a stress level considerably below the strength of fibers, followed by kinking of fibers [3,4]. This mechanism results in premature failure of unidirectional fiber reinforced composites, based on which models were developed in recent decades to predict their compressive strength [5,6,7,8,9]. The proposed models indicate that the microbuckling of fibers is dominated by the shear properties of polymer matrix. Therefore, matrix-tuning becomes an effective approach for enhancing the compressive strength of unidirectional fiber reinforced composites.

In recent years, various nano-fillers, including spherical nanoparticles [10,11], nano-clay [12], graphite nanoplatelets [13], and carbon nanotubes [14] have been incorporated into matrices to fabricate multiscale phase reinforced composites with an improved compressive performance. The results demonstrate that the matrix-tuning by nano-fillers works well for the unidirectional composites based on carbon fibers [13], glass fibers [10,11,12], and basalt fibers [14]. For instance, Sun and his colleagues introduced silica nanoparticles (15 wt.%) [10] and nanoclay (8 wt.%) [12] into epoxy matrices of glass fiber reinforced composites, which led to up to 62% and 36% increases in the compressive strength of the two kinds of composites, respectively. Tsai and Cheng [11] found that by dispersing 30 wt.% silica nanoparticles into epoxy matrix, the compressive strength of glass fiber reinforced composites could be increased by 21% and 16% in quasi-static and dynamic compression tests, respectively. The reinforcing effect of carbon nanotubes with a high aspect ratio seems to be more significant. The addition of 0.4 wt.% multi-walled carbon nanaotubes into epoxy matrix contributes to a 41% increase in the compressive strength of basalt fiber reinforced composites [14].

It is noteworthy that the compressive stress-strain curves of nanocomposite matrices obtained in experiments are essential when applying the classic elastic-plastic microbuckling model, as they limit its prediction capability. This paper provides a solution by developing a method based on loading-unloading tests to establish the constitutive relation of nanocomposite matrices under compression. In combination with the established constitutive relation, a modified microbuckling model can be used to predict the compressive strength of multiscale phase reinforced composites according to a small number of compression tests of nanocomposite matrices. The effectiveness of the proposed model is verified using the experimental data of carbon fiber reinforced composites with epoxy matrices containing uniformly dispersed silica nanoparticles (CF/Nano-SiO_2_/Epoxy composites).

## 2. Modeling

### 2.1. Microbuckling Model with Plasticity and Damage of Polymer Matrix

In the classic elastic-plastic microbuckling model, the compressive strength of unidirectional fiber reinforced composite $\sigma_{c}$ can be expressed as a function of the tangent shear modulus of matrix $G_{m}^{ep}$ and the volume fraction of fibers $v_{f}$ [7]:(1)$$\sigma_{c}=\frac{G_{m}^{ep}}{1-v_{f}},$$
$G_{m}^{ep}$ is estimated by the ratio of incremental shear stress $d\tau_{m}$ to the incremental shear strain $d\gamma_{m}$, i.e., $G_{m}^{ep}=\frac{d\tau_{m}}{d\gamma_{m}}=\frac{d\tau_{m}}{d\gamma_{m}^{e}+d\gamma_{m}^{p}}$. The elastic increment $d\gamma_{m}^{e}$ can be simply calculated through $d\gamma_{m}^{e}=\frac{d\tau_{m}}{G_{m}}$ (where $G_{m}$ represents the shear modulus of matrix). The plastic increment $d\gamma_{m}^{p}$ was considered to be associated with the plasticity of the matrix.

However, experimental study has shown that epoxy matrix demonstrated clear damage during the plastic stage of compression (as discussed in Section 4.1). Thus, the development of damage within the matrix should be included in the model, i.e.,
(2)$$\sigma_{c}=\frac{G_{m}^{epd}}{1-v_{f}},$$
where $G_{m}^{epd}$ is the new tangent shear modulus of matrix influenced by the plasticity and the damage.

To keep this section concise, only key steps and conclusions are provided as follows. The detailed derivation of the modified microbuckling model is provided in the Appendix A section.

Considering that the elastic deformation and plastic deformation are independent from each other, $d\gamma_{m}^{e}$ can be approximated by:(3)$$d\gamma_{12}^{e}=\frac{2(1+\mu )}{E_{m0}(1-d)}d\sigma_{12},$$
where $E_{m0}$ is the initial elastic modulus of the matrix, d represents the damage factor of matrix during compression, and μ is the Poisson’s ratio of the matrix. The relation between $d\gamma_{12}^{p}$ and $d\sigma_{12}$ can be written as:(4)$$d\gamma_{12}^{p}=\frac{3\beta \left(\gamma \beta +3\sin^{2}\theta \right)}{\left(\beta^{2}+3\tan^{2}\theta \right)E_{m}^{p}}d\sigma_{12},$$
where θ is the off-axis angle of fibers in compression tests, as illustrated in Figure 1. $E_{m}^{p}=\frac{d\bar{\sigma }}{d{\bar{\varepsilon }}^{p}}$ is the instantaneous plastic modulus of matrix ($d\bar{\sigma }$ and $d{\bar{\varepsilon }}^{p}$ are the increments of effective stress and effective plastic strain, respectively). Parameters $\beta ={\left(\frac{E_{f}}{E_{m}^{s}}\nu_{f}+\nu_{m}\right)}^{-1}$ and $\gamma ={\left(\frac{E_{f}}{E_{mt}}\nu_{f}+\nu_{m}\right)}^{-1}$, in which $E_{f}$ is the elastic modulus of fiber, $E_{m}^{s}=\frac{\sigma_{11}^{m}}{\varepsilon_{11}^{m}}$ is the secant modulus of a matrix, and $E_{m}^{t}=\frac{d\sigma_{11}^{}}{d\varepsilon_{11}^{}}$ is the tangent modulus of a matrix, which can be estimated as:(5)$$E_{m}^{t}={\left[\frac{1}{E_{m0}(1-d)}+\frac{\beta \left(\gamma \beta +3\sin^{2}\theta \right)}{\gamma \left(\beta^{2}+3\tan^{2}\theta \right)E_{m}^{p}}\right]}^{-1}.$$

Therefore, $G_{m}^{epd}$ can be expressed as:(6)$$G_{m}^{epd}={\left[\frac{2(1+\mu )}{E_{m0}(1-d)}+\frac{3\beta \left(\gamma \beta +3\sin^{2}\theta \right)}{\left(\beta^{2}+3\tan^{2}\theta \right)E_{m}^{p}}\right]}^{-1}$$

Combining with Equations (5) and (6), Equation (2) can be solved numerically through an iterative procedure to attain the critical compressive strength in the case of a specific off-axis angle θ. It was found that the compressive strength of unidirectional fiber reinforced composites demonstrates a quasi-linear relation with the maximum shear stress in matrix caused by the off-axial compression [10,12]. Thus, the longitudinal compressive strength (when the maximum shear stress in matrix vanishes, i.e., $\sigma_{11}$ when $\sigma_{12}=0$) can be extrapolated from the data obtained from the compression tests with a couple of off-axis angles.

### 2.2. Constitutive Model of Nanocomposite Matrix

The solution procedure of microbuckling model needs the instantaneous stress-strain relation of the matrix and its gradient in each iteration step, which is currently acquired from the experimentally recorded curves of the matrix. To break this limit and enhance the prediction ability of the model, a constitutive model of nanocomposite matrices with nanoparticle content in a certain range was developed.

In addition to the non-zero residual strain obtained after unloading the epoxy specimen in the plastic deformation stage, a reduction of the modulus was observed during the unloading process in our experimental study (see Section 4 for an example), indicating the development of damage within the matrix. Therefore, both the damage and plastic behavior contributed to the nonlinear stress-strain relation of the epoxy matrix. According to the Ladeveze damage model of composite laminates [15], which can be simplified into a form for isotropic materials, the nonlinear stress-strain relation of a matrix under compression can be established through loading-unloading experiments.

From the viewpoint of damage, the nominal damage strain energy of matrix Y can be approximated as a function of the elastic strain $\varepsilon_{i}^{e}$ at the beginning of the ith unloading:(7)$$Y(\varepsilon_{i})=\sqrt{\frac{1}{2}E_{0}{\left(\varepsilon_{i}^{e}\right)}^{2}}$$
where $E_{0}$ is the initial modulus of the matrix. Y can also be expressed in another form, as a function of the damage factor of the matrix at *i*th unloading $d_{i}$:(8)$$Y(d_{i})=Y_{0}+Y_{C}d_{i}$$
where $d_{i}$ can be estimated through $d_{i}=1-\frac{E_{i}}{E_{0}}$. $Y_{0}$ denotes the threshold of the initial damage. It is assumed that no damage occurs in the matrix (i.e., d=0) if $Y(\varepsilon_{i})<Y_{0}$. Above this threshold, damage appears in the matrix and gradually develops along with the increasing strain. $Y_{C}$ represents the threshold of the damage. According to Equation (8), the larger $Y_{C}$ is, the slower the damage factor $d_{i}$ increases. Thus, the stress-strain relation at the beginning of ith unloading can be expressed as:(9)$$\sigma_{i}=E_{i}\varepsilon_{i}^{e}=E_{0}\left(1-d_{i}\right)\varepsilon_{i}^{e}=\frac{E_{0}\varepsilon_{i}^{e}}{Y_{C}}\left(Y_{C}+Y_{0}-\sqrt{\frac{1}{2}E_{0}{\left(\varepsilon_{i}^{e}\right)}^{2}}\right).$$

To introduce the influence of the matrix plasticity, the relation between the stress $\sigma_{i}$ and plastic strain $\varepsilon_{i}^{p}$ can be constructed based on the modified Lukwik equation [16]:(10)$$\frac{\sigma_{i}}{\left(1-d_{i}\right)}-\left[\sigma_{0}+h\cdot {\left(\varepsilon_{i}^{p}\right)}^{m}\right]=0$$
where $\sigma_{0}$ is the initial yield stress, while *h* and *m* denote the strength coefficient and work-hardening exponent, respectively.

While assuming that there is no coupling of the elastic strain and the plastic stain, at the beginning of the *i*th unloading the total strain satisfies:(11)$$\varepsilon_{i}=\varepsilon_{i}^{e}+\varepsilon_{i}^{p}.$$

The initial modulus $E_{0}$ of a nanocomposite matrix varies with the volume fraction of nanoparticles $v_{p}$. It can be estimated by the Mori-Tanaka method [17], treating the Nano-SiO_2_ particles as spherical inclusions:(12)$$\begin{array}{l} G_{0}=G_{m}+\frac{G_{p}-G_{m}}{1+4\left(1-v_{p}\right)G_{H}\left(G_{p}-G_{m}\right)}v_{p}\\ G_{H}=\frac{3\left(2G_{m}+K_{m}\right)}{10G_{m}\left(4G_{m}+3K_{m}\right)} \end{array}$$
where $G_{0}$, $G_{m}$, and $G_{p}$ represent the shear moduli of the nanocomposite, epoxy matrix, and particle inclusion, respectively. $K_{m}$ is the bulk modulus of the matrix. As the nanocomposite matrix can be regarded as an isotropic material, these moduli satisfy:(13)$$\begin{array}{l} \begin{array}{ccc} G_{0}=\frac{E_{0}}{2\left(1+\mu_{0}\right)}, & G_{m}=\frac{E_{m}}{2\left(1+\mu_{m}\right)},\; & G_{p}=\frac{E_{p}}{2\left(1+\mu_{p}\right)} \end{array}\\ K_{m}=\frac{E_{m}}{3\left(1-\mu_{m}\right)} \end{array}$$
where $\mu_{0}$, $\mu_{m}$, and $\mu_{p}$ are the Poisson’s ratios of nanocomposite matrix, epoxy resin, and silica respectively. Taking approximately $\mu_{0}\approx \mu_{m}\approx 0.33$ and $\mu_{p}\approx 0.16$, the modulus of Nano-SiO_2_/Epoxy matrix can be predicted by:(14)$$E_{0}=E_{m}+\frac{2.66v_{p}\left(2.32E_{p}-2.66E_{m}\right)}{1+1.24\left(1-v_{p}\right)\left(2.32E_{p}-2.66E_{m}\right)/E_{m}}.$$

## 3. Experimental Verification

### 3.1. Materials and Sample Preparation

Unidirectional fiber reinforced composites were prepared using Toray T700 carbon fiber sheets. Each fiber bundle in the cloth contains approximately 12,000 monofilaments. The longitudinal elastic modulus of carbon fibers is about 240 GPa [18]. The epoxy resins include two commercial types: (i) Bisphenol A (BPA) epoxy resin (E51) with an epoxide equivalent of 196 g/mol, supplied by China National BlueStar Co. Ltd. (Beijing, China); (ii) Bisphenol F (BPF) epoxy resin (Nanopox F520) with an epoxy equivalent of 277 g/mol, supplied by Evonik Industries AG (Essen, Germany). The Nanopox F520 resin, containing 40 wt.% SiO_2_ nanoparticles with an average diameter of 25 nm, was synthesized and incorporated through the sol-gel technology, which guarantees a very uniform dispersion [19]. It was diluted with E51 resin to prepare nanocomposite matrices with various volume fractions of nanoparticles. The curing agent is methylhexahydrophthalic anhydride (MHHPA) with a molecular weight of 168 g/mol, supplied by Puyang Huicheng Electronic Material Co. Ltd. (Puyang, China). N, N-Dimethylbenzylamine (BDMA) from Sinopharm Chemical Reagent Co. Ltd. (Shanghai, China) serves as an accelerator.

Table 1 lists the recipes of the nanocomposite matrices prepared in this study. The volume fraction of Nano-SiO_2_ was controlled in a range from 0 vol.% to 8.7 vol.% (i.e., 0 wt.%–15 wt%). All the materials were dried at 60 °C for 6 h prior to sample preparation. The two kinds of resins, curing agent, and accelerator were mixed using mechanical stirring for about 20 min. Then the mixture was kept in a vacuum chamber for 30 min for degas. Afterwards, it was cast into silicone rubber molds to make 80 mm-long dumbbell-shape specimens for tension (ISO 527) and cubic specimens with a dimension of 12 mm × 12 mm × 20 mm for compression (ISO 604). The curing process followed a three-step procedure: (i) the mixture was kept at 60 °C in an oven for 480 min; (i) then the temperature was raised to 100 °C and kept for 120 min; (iii) the pre-cured specimens were removed from the molds and finished their post-curing at 150 °C for 300 min.

Unidirectional CF/Nano-SiO_2_/Epoxy composites were prepared through the vacuum assisted resin transfer molding method. The mixture of resins, hardener, and accelerator were injected into the mold, on which 32 piles of carbon fiber sheets were stacked. The curing process follows same procedure mentioned above. Figure 1b shows the block specimens cut from the prepared laminates for longitudinal compression test. The off-axial angles of fibers were set as 2°, 5°, 10°, 15°, and 20°.

### 3.2. Characterization and Mechanical Tests

Figure 2a shows the micrographs of cryo-fractured surface of epoxy matrix with 6.0 vol.% Nano-SiO_2_, which was taken on a scanning electron microscope (ZEISS SIGMA 300, Oberkochen, Germany). The nanoparticles were dispersed very uniformly in an epoxy matrix without visible agglomeration.

Figure 2b shows the microscopic images of the cross-section and lateral side of the prepared laminates respectively, taken on an Olympus BX51 microscope (Tokyo, Japan). It can be observed that the carbon fibers are uniformly distributed and well impregnated with resins. The average diameter and volume fraction of fibers are measured to be about 7 μm and 42 vol.%.

Tensile tests and compression tests carried out on an Instron 5567 universal testing machine (Norwood, MA, USA). The strain rate of deformation was kept at $10^{-4}/s$. For each composite system (five volume fractions of Nano-SiO_2_), 5–8 valid data were collected under each off-axial angle (five angles) to offset the influence of accidental error. It is noteworthy that only four nanocomposite matrices (with Nano-SiO_2_ contents of 0 vol.%, 1.1 vol.%, 2.8 vol.%, and 6.0 vol.%) were tested. The compressive stress-strain relation of epoxy matrix with 8.7 vol.% Nano-SiO_2_ can be predicted by the proposed model and fed to the microbuckling model. Finally, the predicted compressive strength of unidirectional composite using this matrix is compared with the measured value.

## 4. Results and Discussion

### 4.1. Compressive Stress-Strain Relation of Nano-SiO_2_/Epoxy Matrices

Table 2 lists the elastic moduli of epoxy matrices with various nanoparticle contents measured in tensile tests. It can be seen that the stiffness of epoxy matrices was improved by the incorporation of nanoparticles to some extent. 6.0 vol.% Nano-SiO_2_ leads to a 13.5% increase in the modulus of the epoxy matrix. The moduli of nanocomposite matrices can be predicted by substituting the modulus of silica (77 GPa [20]) and the measured elastic modulus of neat epoxy matrix into Equation (14), as also listed in Table 2. The predicted values are slightly lower than the measured values with deviations of up to 3%.

Figure 3 demonstrates the stress-strain curve recorded in the loading-unloading process for neat epoxy matrix, according to which $Y_{0}$, $Y_{C}$, $\sigma_{0}$, h, and m in Equations (8) and (10) were estimated by fitting. It can be observed that the modulus of epoxy matrix gradually decreased with the increasing stress level at the start points of the unloading process, indicating the occurrence of damage within the epoxy matrix. Moreover, non-zero residual strain can be found when the applied loading was completely removed, confirming the occurrence of non-negligible plastic deformation. It is noteworthy that five cycles were performed in our study (as also reported in the literature using the Ladeveze damage model [15,21,22]), though theoretically only two cycles were required to obtain $Y_{0}$ and $Y_{C}$, and three cycles were required to get $\sigma_{0}$, h, and m. This configuration was based on the following considerations: (i) more data points helps to minimize the disturbance of accidental error in the experiments; (ii) the five unloading points were set at different stress levels (80 MPa, 90 MPa, 100 MPa, 105 MPa, and 108 MP in our experiments), which can be used to validate whether the linear relation between Y and d holds during the stable deformation stage of the epoxy matrix.

Figure 4 shows the fit lines of $Y\left(d_{i}\right)$ and $d_{i}$ for the four composite systems. The coefficients of determination ($R^{2}$) of all fit lines were above 0.966, which confirms the linear relation described in Equation (8). Figure 5 shows the fitting of the parameters in Equations (8) and (10) for the four composite systems with various Nano-SiO_2_ contents. As no proper physical models are currently available to describe the observed relation between these parameters and Nano-SiO_2_ volume fraction, the following empirical equations were established:(15)$$\begin{array}{l} Y_{0}\left(v_{p}\right)=-4.6\times 10^{-5}v_{p}+8.8\times 10^{-4}\\ Y_{C}\left(v_{p}\right)=4.35\times 10^{-4}\times \ln\left(v_{p}+1\right)+3.05\times 10^{-3} \end{array}$$
(16)$$\begin{array}{l} \sigma_{0}(v_{p})=4.64\times 10^{-3}v_{p}+0.0424\\ h(v_{p})=4.91\times 10^{-2}v_{p}+0.3024\\ m(v_{p})=1.618\times 10^{-2}v_{p}+0.128 \end{array}$$

It was found that most parameters except $Y_{C}$ demonstrated a quasi-linear relation with Nano-SiO_2_ content. Table 3 lists the values of these parameters at specific Nano-SiO_2_ volume fractions. The values of parameters in the case of $v_{p}=8.7$ vol.% are extrapolated from the curves constructed according to Equations (15) and (16), which are also plotted as hollow marks in Figure 5.

Figure 6 compares the predicted compressive stress-strain curves of nanocomposite matrices with the measured data. The prediction reaches a good agreement with the experimental results. In particular, the relative deviation between the stress-strain curves of epoxy matrix with 8.7 vol.% Nano-SiO_2_ obtained by the proposed model and by the experiments falls within a range up to 8%.

### 4.2. Compressive Strength of CF/Nano-SiO_2_/Epoxy Composites

Table 4 lists the measured compressive strength of multiscale composites with various off-axial angles. The maximum compressive stress and maximum shear stress can be extracted from the measured off-axial compressive strength, which demonstrate a quasi-linear relation, as shown in Figure 7. The longitudinal compressive strength of composites can be approximated by extrapolation of the fitting lines. It can be found that the modification of epoxy matrix with stiff nanoparticles significantly improve the compressive performance of the unidirectional fiber reinforced composites. An increase of 62.7% was achieved by the incorporation of 8.7 vol.% Nano-SiO_2_.

Figure 8a compares the off-axial compressive strength predicted by the microbuckling model using the experimental stress-strain curves of matrices and the curves of constitutive model. The two sets of data are quite close to each other, with relative deviation below 8%. Although the predicted stresses at some Nano-SiO_2_ contents became higher than the measured values near the compressive strength (Figure 6), this deviation had trivial effects on the prediction of the microbuckling model because the stress within the epoxy matrix was still far below its compressive strength when the unidirectional fiber reinforced composites failed under off-axial compression.

Figure 8b gives the longitudinal compressive strength of CF/Nano-SiO_2_/Epoxy composites estimated by extrapolation from the experimental data and predicted data in Figure 8a, which demonstrates an ascendant tendency with the increase of Nano-SiO_2_ content. The two data sets also agreed well with each other. The relative deviation was up to 8.2%, which still falls in the range of standard deviation obtained in experiments (about 4%–10%). In particular, the relative deviation of the predicted compressive strength at 8.7 vol.% Nano-SiO_2_ achieved a low level of about 2%.

## 5. Conclusions

The compressive strength of unidirectional carbon fiber reinforced composites can be significantly improved by the incorporation of uniformly dispersed SiO_2_ nanoparticles into epoxy matrix. The reinforcing effects of nanoparticles was accurately described by importing the constitutive model of a nanocomposite matrix under compression into a modified elastic-plastic microbuckling model, including the influence of matrix damage. The prediction of the proposed models achieved an excellent agreement with the experimental results. Based on the approach developed in this study, the compressive strength of multiscale composites can be efficiently evaluated by using a small number of material tests. The model may provide a basis for estimating the properties of advanced composites for the designers and analyzers working on the performance of composite parts with complicated structures and under practical loading conditions, when they assign the material properties to finite element models.

## Figures and Tables

**Figure 1 materials-12-03897-f001:**
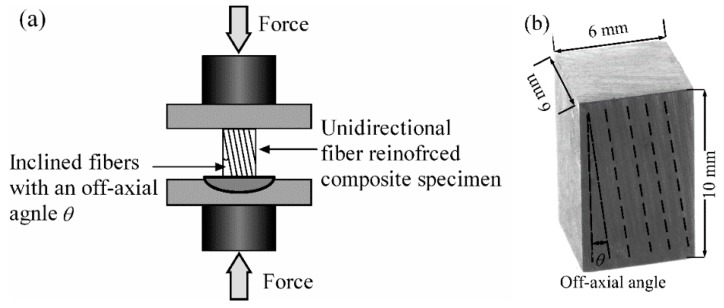
(**a**) Setup of off-axial compression test for unidirectional fiber reinforced composites and (**b**) Dimension of the prepared composite specimen.

**Figure 2 materials-12-03897-f002:**
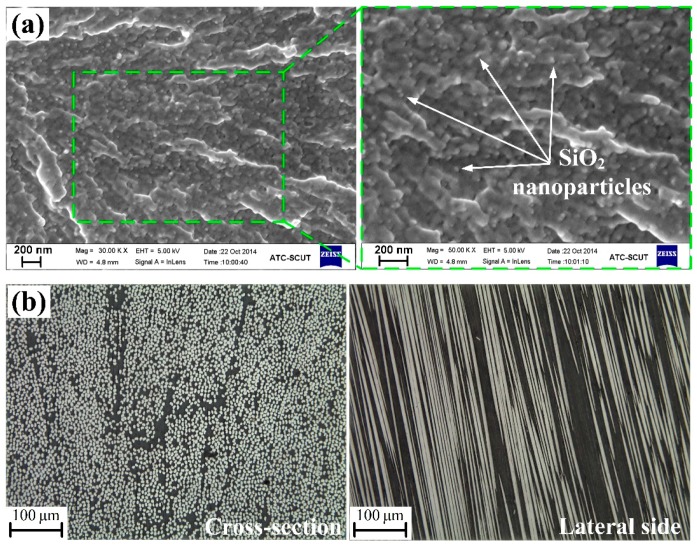
(**a**) SEM micrograph of cryo-fractured surface of epoxy matrix with 6.0 vol.% Nano-SiO_2_ and (**b**) Optical micrographs of cross-section and lateral side of unidirectional carbon fiber reinforced composites.

**Figure 3 materials-12-03897-f003:**
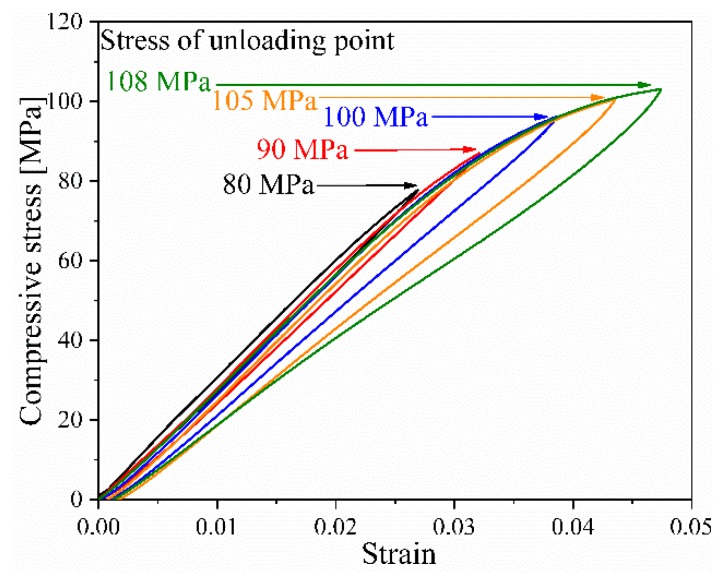
Loading-unloading curves of a neat epoxy matrix during compression.

**Figure 4 materials-12-03897-f004:**
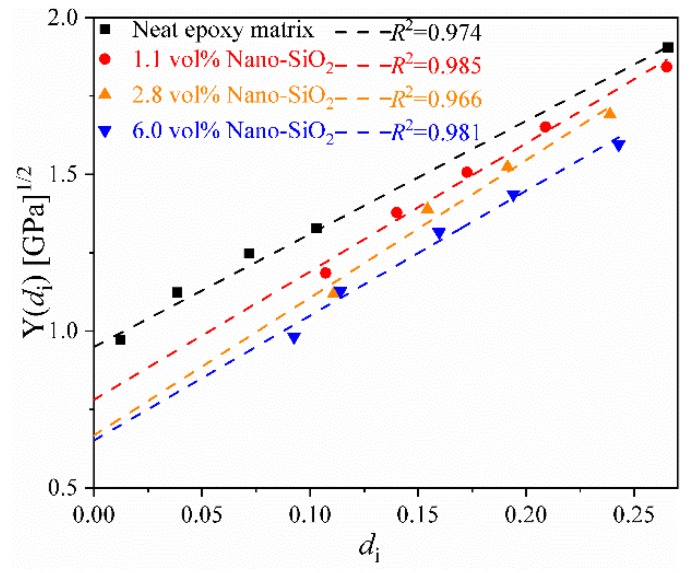
$Y-d_{i}$ relation of nanocomposite matrices during the loading-unloading cycles.

**Figure 5 materials-12-03897-f005:**
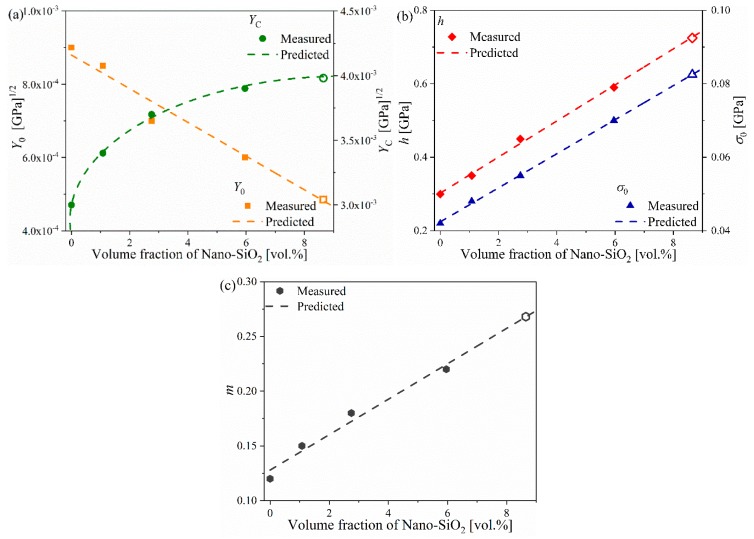
Variation of (**a**) $Y_{0}$, $Y_{C}$, (**b**) $\sigma_{0}$, h, and (**c**) m versus volume fraction of Nano-SiO_2_ ($v_{p}$).

**Figure 6 materials-12-03897-f006:**
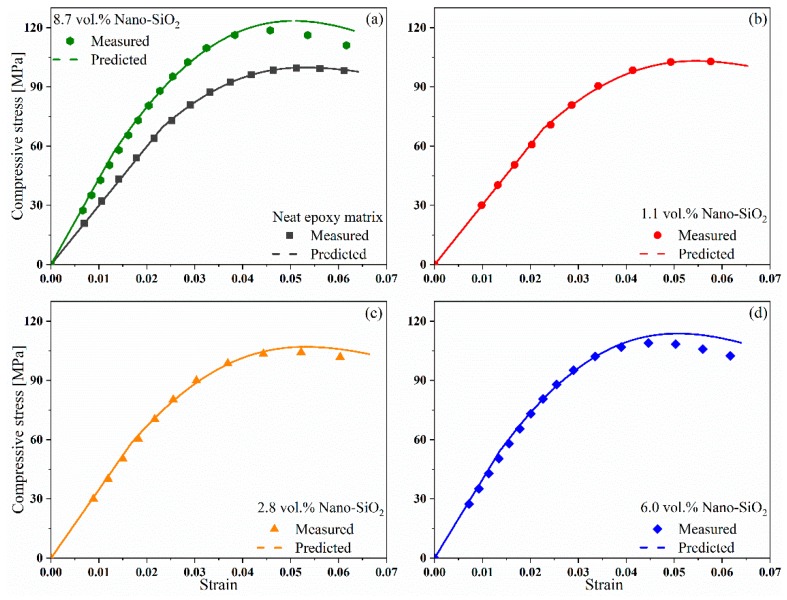
Predicted compressive stress-strain curves of Nano-SiO_2_/Epoxy matrices in comparison with measured data points. (**a**) composite with neat epoxy matrix; (**b**) composite with 1.1 vol% Nano-SiO_2_ modified epoxy matrix; (**c**) composite with 2.8 vol% Nano-SiO_2_ modified epoxy matrix; (**d**) composite with 6.0 vol% Nano-SiO_2_ modified epoxy matrix.

**Figure 7 materials-12-03897-f007:**
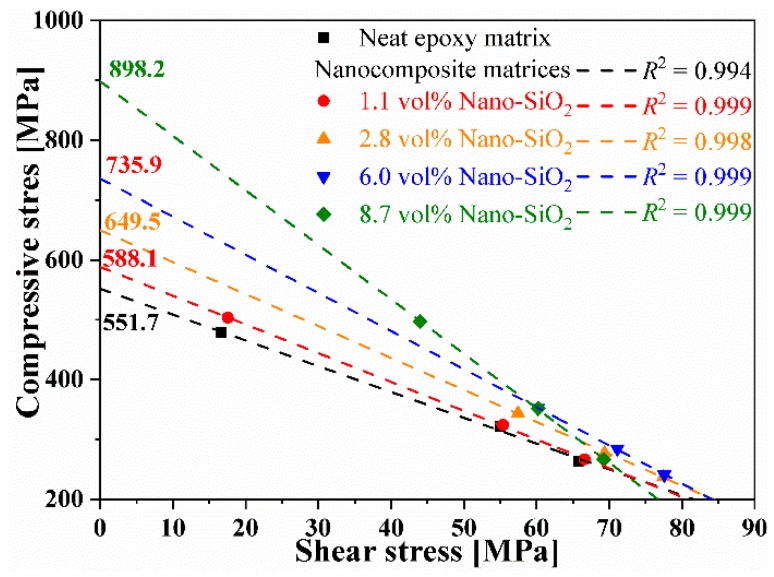
Maximum compressive stress and shear stress extracted from the measured off-axial compressive strength of CF/Nano-SiO_2_/Epoxy composites.

**Figure 8 materials-12-03897-f008:**
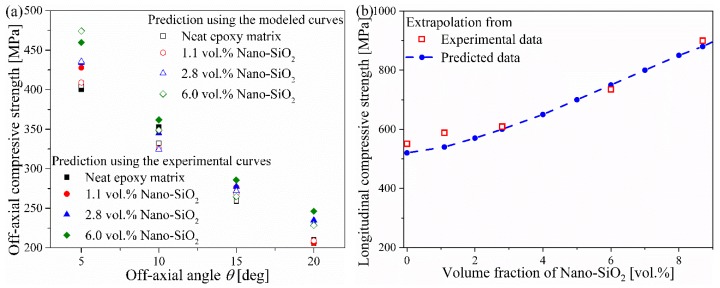
(**a**) Off-axial compressive strength and (**b**) extrapolated longitudinal compressive strength of CF/Nano-SiO_2_/Epoxy composites.

**Table 1 materials-12-03897-t001:** Recipes of the epoxy matrices with various contents of nanoparticles.

Nano-SiO_2_ [vol.%/wt.%]	E51 [Mass Part]	Nanopox F520 [Mass Part]	MHHPA [Mass Part]	BDMA [Mass Part]
0/0	100	0	85.9	1.00
1.1/2	100	10.1	91.9	1.10
2.8/5	100	29.0	103.3	1.20
6.0/10	100	77.6	132.0	1.55
8.7/15	100	177.7	196.1	2.54

**Table 2 materials-12-03897-t002:** Measured and predicted elastic moduli of Nano-SiO_2_/Epoxy matrices.

Nano-SiO_2_ [vol.%]	Elastic Modulus [GPa]	Relative Improvement [%]	Predicted Modulus $E_{0}$ [GPa]
0	3.25 ± 0.28		
1.1	3.37 ± 0.11	3.7	3.32
2.8	3.51 ± 0.21	8.0	3.44
6.0	3.69 ± 0.33	13.5	3.68
8.7			3.87

**Table 3 materials-12-03897-t003:** Values of material characteristics estimated using Equations (15) and (16).

Nano-SiO_2_ [vol.%]	$Y_{0}^{}$ [GPa]^1/2^	$Y_{C}^{}$ [GPa]^1/2^	$\sigma_{0}^{}$ [GPa]	$h_{}^{}$ [GPa]	*m*
0	$9.0\times 10^{-4}$	$3.0\times 10^{-3}$	0.042	0.31	0.12
1.1	$8.5\times 10^{-4}$	$3.4\times 10^{-3}$	0.048	0.35	0.15
2.8	$7.0\times 10^{-4}$	$3.7\times 10^{-3}$	0.055	0.45	0.18
6.0	$6.0\times 10^{-4}$	$3.9\times 10^{-3}$	0.071	0.59	0.22
8.7	$4.7\times 10^{-4}$	$4.0\times 10^{-3}$	0.083	0.73	0.27

**Table 4 materials-12-03897-t004:** Off-axial compressive strength of CF/Nano-SiO_2_/Epoxy composites.

Nano-SiO_2_ [vol.%]	Off-Axial Compressive Strength [MPa]				
	2°	5°	10°	15°	20°
0		386.3 ± 10.4	319.0 ± 20.3	263.9 ± 11.8	202.2 ± 28.9
1.1	503.4 ± 47.5	391.2 ± 33.9	324.1 ± 23.6	266.5 ± 17.7	190.8 ± 6.4
2.8	514.8 ± 16.3	403.8 ± 20.1	339.1 ± 17.9	280.5 ± 13.6	240.8 ± 20.6
6.0	547.4 ± 24.3	416 ± 21.4	347.9 ± 17.7	286.8 ± 18.8	241.3 ± 9.8
8.7	569.9 ± 22.5		352.1 ± 19.2	266.9 ± 16.5	219.3 ± 11.5

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
