# Peer review of "Modeling of Compressive Strength for Unidirectional Fiber Reinforced Composites with Nanoparticle Modified Epoxy Matrix"

_materials, 2019, doi:10.3390/ma12233897_

Round 1

Reviewer 1 Report

The manuscript is relevant in its field, deals with an interesting issue and is well explained and. However, there are certain minor changes required.

The scale of each picture of Figure 2 should be clearly visible.

In Figure 3 it is recommended to include markers in the curves plotted.

Figure 4 should include the R2 values of the fitting performed. A analogous comment can be carried out regarding Figure 7

Author Response

Comment: The manuscript is relevant in its field, deals with an interesting issue and is well explained and. However, there are certain minor changes required.

Response: We appreciate the reviewer’s positive comment on our paper. The paper has been carefully revised according to the suggestions.

Comment: The scale of each picture of Figure 2 should be clearly visible.

Response: We modified Figure 2 to make the scales clearer.

Comment: In Figure 3 it is recommended to include markers in the curves plotted.

Response: We understand the curves may be indistinguishable from each other when they appear in a printed greyscale version. Thus, we modified the annotations and added some arrows to make Figure 3 more self-explanatory.

Comment: Figure 4 should include the R2 values of the fitting performed. A analogous comment can be carried out regarding Figure 7.

Response: We added the R-squared values of fit lines into Figure 4 and Figure 7, as suggested by the reviewer.

Reviewer 2 Report

The paper is generally well written and organized.

However, revision is required before its final publication:

"fortunately": remove this term, and other subjective expressions from the text. Prefer objective expressions only for the discussion of results the main topic of the paper is the compressive behaviour of tested samples. In civil engineering practice, however, recent applications proved the benefits of fiber composites when are exploited for their high tensile resistance. Like for example in this study: https://www.sciencedirect.com/science/article/pii/S235271021830771X. Do the authors expect some sensitivity also for the tensile resistance? Can the studied samples typology possibly used for applications like the mentioned example? for fiber composites in civil engineering (or other fields) it would be beneficial for the reader to add in the paper some brief comments (a couple of sentences) to show where this compressive behaviour can be crucial (see also my previous comment) table 1: it would be better to mention the number of samples for the calculation of the values collected in the table (average) the same comment is for table 2 there's a very good fitting between predicted and measured data, in the section of experimental results. How these values can be generalized? Can these comparisons be sensitive to the loading conditions? Thus "ideal" loading conditions of the experimental setup, compared to "real" applications and boundaries? Please add some comments

Author Response

Comment: The paper is generally well written and organized. However, revision is required before its final publication:

Response: We appreciate the reviewer’s positive comment on our paper. The paper has been carefully revised according to the suggestions. All the modified parts are highlighted with yellow color.

Comment: "fortunately": remove this term, and other subjective expressions from the text. Prefer objective expressions only for the discussion of results.

Response: We modified the expression and deleted this word. In addition, we check the paper thoroughly for this issue.

Comment: The main topic of the paper is the compressive behaviour of tested samples. In civil engineering practice, however, recent applications proved the benefits of fiber composites when are exploited for their high tensile resistance. Like for example in this study: https://www.sciencedirect.com/science/article/pii/S235271021830771X. Do the authors expect some sensitivity also for the tensile resistance? Can the studied samples typology possibly used for applications like the mentioned example? For fiber composites in civil engineering (or other fields) it would be beneficial for the reader to add in the paper some brief comments (a couple of sentences) to show where this compressive behaviour can be crucial (see also my previous comment).

Response: Fiber reinforced composites are generally designed to carry the tensile load along the fibers because of their superior modulus and strength under that loading conditions. However, fibers with high aspect ratio show significant instability when they are subjected to longitudinal compression, which makes the compressive performance of fiber reinforced composites a different issue.

In practical applications, fiber reinforced composite components inevitably experience the longitudinal compression because their deformation in service may be quite complex. Those components may be accidentally bent, when part of the fibers is under longitudinal compression. Therefore, consideration of compressive properties for fiber reinforced composites is important to the safe and optimal design of components.

Our study demonstrates the methods to enhance and estimate the compressive strength of unidirectional fiber reinforced composites with nanoparticle-modified matrix, which may help the engineers in the relevant areas to improve their design of composite parts.

We would thank the reviewers for sharing their points and an inspiring reference paper. We learned this paper and cite it in the revised manuscript along with the statement about the issues mentioned above (Section 1 and 5).

Comment: Table 1: it would be better to mention the number of samples for the calculation of the values collected in the table (average) the same comment is for table 2.

Response: For each composite system (five volume fractions of Nano-SiO2), 5 – 8 valid data were collected under each off-axial angle (five angles) to offset the influence of accidental error.

We added this statement into the description of experiments (Section 3.2).

Comment: There's a very good fitting between predicted and measured data, in the section of experimental results. How these values can be generalized? Can these comparisons be sensitive to the loading conditions? Thus "ideal" loading conditions of the experimental setup, compared to "real" applications and boundaries? Please add some comments.

Response: As the testing configuration used in this study is a popular one (see Refs. [6, 9-11]), we are confident that our model can be employed by the other researchers to analyze the compressive performance of fiber reinforced composites with the matrix modified by other nanofillers.

To those working on the performance of composites with more complicated structures and under practical loading conditions, our study may provide them a solid basis to estimate the properties of advanced composites, e.g. when they assign the material properties to finite element models.

We added the statement about this issue into Section 5.